# The Ubiquitin-Proteasome System Participates in Sperm Surface Subproteome Remodeling during Boar Sperm Capacitation

**DOI:** 10.3390/biom13060996

**Published:** 2023-06-15

**Authors:** Michal Zigo, Karl Kerns, Peter Sutovsky

**Affiliations:** 1Division of Animal Science, University of Missouri, Columbia, MO 65211, USA; kkerns@iastate.edu; 2Department of Animal Science, Iowa State University, Ames, IA 50011, USA; sutovskyp@missouri.edu; 3Department of Obstetrics, Gynecology and Women’s Health, University of Missouri, Columbia, MO 65211, USA

**Keywords:** ubiquitin–proteasome system, spermatozoon, sperm capacitation, sperm surface protein, CSE1L, PFDN4, STK39/SPAK, pig

## Abstract

Sperm capacitation is a complex process endowing biological and biochemical changes to a spermatozoon for a successful encounter with an oocyte. The present study focused on the role of the ubiquitin–proteasome system (UPS) in the remodeling of the sperm surface subproteome. The sperm surface subproteome from non-capacitated and in vitro capacitated (IVC) porcine spermatozoa, with and without proteasomal inhibition, was selectively isolated. The purified sperm surface subproteome was analyzed using high-resolution, quantitative liquid chromatography–mass spectrometry (LC-MS) in four replicates. We identified 1680 HUGO annotated proteins, out of which we found 91 to be at least 1.5× less abundant (*p* < 0.05) and 141 to be at least 1.5× more abundant (*p* < 0.05) on the surface of IVC spermatozoa. These proteins were associated with sperm capacitation, hyperactivation, metabolism, acrosomal exocytosis, and fertilization. Abundances of 14 proteins were found to be significantly different (*p* < 0.05), exceeding a 1.5-fold abundance between the proteasomally inhibited (100 µM MG132) and vehicle control (0.2% ethanol) groups. The proteins NIF3L1, CSE1L, NDUFB7, PGLS, PPP4C, STK39, and TPRG1L were found to be more abundant; while BPHL, GSN, GSPT1, PFDN4, STYXL1, TIMM10, and UBXN4 were found to be less abundant in proteasomally inhibited IVC spermatozoa. Despite the UPS having a narrow range of targets, it modulated sperm metabolism and binding by regulating susceptible surface proteins. Changes in CSE1L, PFDN4, and STK39 during in vitro capacitation were confirmed using immunocytochemistry, image-based flow cytometry, and Western blotting. The results confirmed the active participation of the UPS in the extensive sperm surface proteome remodeling that occurs during boar sperm capacitation. This work will help us to identify new pharmacological mechanisms to positively or negatively modulate sperm fertilizing ability in food animals and humans.

## 1. Introduction

Protein ubiquitination is a stable, reversible process where a small (76 AA, 8.5 kDa), evolutionarily conserved chaperone protein called ubiquitin is covalently ligated through its Gly76 terminal residue to the ε-NH_2_ group of an internal lysine of the substrate [1]. The same isopeptide bond is formed between the ubiquitin moieties of the polyubiquitin chain, in particular, between G76 and K48 [2], which serves as a proteolysis recognition signal for the endpoint protease, namely, the 26S proteasome [3,4]. The three classes of enzymes that are involved in the cascade reaction of ubiquitin conjugation are (i) E1—the ubiquitin-activating enzymes, (ii) E2—the ubiquitin-conjugating/carrier enzymes, and (iii) E3—the ubiquitin protein ligases [5,6]. The 26S proteasome is a large (∼1.5 MDa), multicatalytic protease that degrades polyubiquitinated proteins into small peptides. The structure of the canonical 26S proteasome, as well as functions of its components, can be found in Ciechanover et al. [7] and Sutovsky [7,8]. Functionally, the 26S proteasome is responsible for cellular proteostasis, i.e., the protein degradation and turnover of misfolded or damaged proteins or proteins at the end of their natural lifespan. Degradation of intracellular proteins via the ubiquitin–proteasome system (UPS) is also involved in the regulation of a variety of cellular processes, such as cell cycle and division, regulation of transcription factors, and assurance of cellular quality control [9,10,11].

Sperm capacitation, observed independently by Austin and Chang in 1951 [12,13] is the terminal sperm maturation process that takes place in the oviductal sperm reservoir [14]. This process is triggered in response to ovulation, leading to a series of biological and biochemical changes in spermatozoa. These include the activation of proton channels and Na^+^/HCO_3_^−^ transporters, resulting in the alkalinization of the cytoplasm and membrane hyperpolarization and activation of pH-sensitive Ca^2+^ channels, such as CatSper [15,16], leading to Ca^2+^ influx. Soluble adenylyl cyclase (sAC, also known as ADCY10) is activated by bicarbonate and calcium ions [17], leading to an increase in cyclic adenosine monophosphate (cAMP), which results in protein tyrosine phosphorylation, which is also a known hallmark of sperm capacitation [18,19]. Slower changes that accompany sperm capacitation are the removal of decapacitating factors and cholesterol [20,21]. We reported capacitation-associated zinc effluxes; however, the dynamics of this capacitation-related change remain unexplored [22]. The capacitation related changes are prerequisites for sperm hyperactivation, aiding spermatozoa to detach from the oviductal epithelium within the sperm reservoir: plasma membrane lipid raft reorganization is facilitated through increased membrane fluidity at the apical ridge region, while membrane docking and acrosomal remodeling occur in preparation for acrosomal exocytosis, zona pellucida penetration, and sperm–oolemma adhesion and fusion [23,24]. In vivo, spermatozoa capacitate asynchronously to continuously replenish the pool of freshly capacitated spermatozoa to maximize the chances of a gamete encounter. Once spermatozoa are irreversibly capacitated, fertilization or sperm death occurs [25].

Our group has dedicated more than twenty years of research effort to understanding the role of the UPS in mammalian reproduction; see pertinent reviews [8,26,27,28,29,30,31,32,33,34,35,36,37,38]. We showed that the UPS is involved in many aspects of mammalian reproduction, starting with gametogenesis and ending with paternal mitophagy and major zygotic genome activation. This intensive focus dedicated to understanding the role of the UPS by our and other groups has borne new knowledge in our understanding of mammalian sperm capacitation. Components of the UPS were shown to participate in capacitation events, such as degradation of protein kinase cAMP-dependent type I regulatory subunit alpha (PRKAR1) and A-kinase anchoring protein 3 (AKAP3) protein [39,40,41]; acrosomal remodeling, protein processing, and compartmentalization [42,43,44]; hyperactivation [41,45]; and spermadhesin de-aggregation from the sperm surface necessary for sperm detachment from the oviductal epithelium [46]. In an in vitro and ex vivo capacitation assay, the 26S proteasome is physiologically important for the detachment of spermatozoa from oviductal glycan beads and explants [47,48]. A recent computational study suggested the 26S proteasome to be one of two key regulators of mammalian sperm capacitation [49]. Further, the proteolytic activity of the 26S proteasome seems to be important during the early stages of sperm capacitation [49,50] and is regulated by PRKA in a feedback loop [51,52].

Sperm surface subproteome remodeling during sperm capacitation is a necessity for spermatozoa to detach from the oviductal epithelium reservoir and recognize zona pellucida glycoproteins. Since proteasomal inhibition prevents sperm detachment from oviductal glycan beads and explants, mimicking oviductal epithelium, we addressed the following question: What sperm surface protein changes does the UPS orchestrate during boar sperm capacitation to alter the sperm binding affinity, allowing spermatozoa to detach from the oviductal epithelium and bind to oocyte zona pellucida? The objective of this study was, therefore, to (i) selectively isolate the sperm surface subproteome from non-capacitated and capacitated spermatozoa with or without proteasomal inhibition using surface biotinylation, (ii) identify targets of the UPS regulation using high-resolution mass spectrometry, and (iii) screen three significantly different UPS target proteins during sperm capacitation by using our protein phenotyping pipeline.

## 2. Materials and Methods

### 2.1. Antibodies and Reagents

Proteasomal inhibitor MG132 (cat # BML-PI102) was purchased from ENZO Life Sciences (Farmingdale, NY, USA). For the biotin-directed affinity purification, a Pierce^TM^ Cell Surface Biotinylation and Isolation kit (cat # A44390) was purchased from ThermoFisher Scientific, Rockford, IL, USA. For indirect immunofluorescence, image-based flow cytometry, and for Western blotting, rabbit polyclonal anti-CSE1L (cat # PA5-86130), rabbit polyclonal anti-PFDN4 (cat # PA5-110100), and rabbit polyclonal anti-STK39 (cat # PA5-17338) antibodies, mouse IgG isotype control (cat # 31903), and normal rabbit serum (cat # 31883) were all purchased from Invitrogen, Carlsbad, CA, USA. Mouse monoclonal anti-STK39 (cat # NBP1-48021) was purchased from Novus Biologicals, Centennial, CO, USA. Mouse monoclonal anti-β-tubulin antibody, clone E7 (Antibody Registry ID: AB_2315513) was bought from the Developmental Studies Hybridoma Bank of the University of Iowa, Iowa City, IA, USA. Goat anti-mouse IgG, AF488 (cat # A11001), goat anti-mouse IgG, HRP (cat # 31430), goat anti-rabbit IgG, AF488, goat anti-rabbit IgG, AF647 (cat # A21244), goat anti-rabbit IgG, HRP (cat # 31460), lectin PNA, AF568 (cat # L32458), and DAPI (cat # D1306) were purchased from Invitrogen, Carlsbad, CA, USA. For Western blotting, the following reagents were used: Halt^TM^ Protease and Phosphatase inhibitor cocktail, EDTA free (cat # 78443) was purchased from ThermoFisher Scientific, Rockford, IL, USA; NuPAGE^TM^ 4-12% Bis-Tris gel (cat # NP0329BOX) and Novex^®^ Sharp Pre-stained Protein Standard were purchased from Invitrogen, Carlsbad, CA, USA; Bradford protein assay dye (cat # 5000006) was purchased from Bio-Rad, Hercules, CA, USA; PVDF Immobilon Transfer Membrane and Luminata Crescendo Western HRP Substrate were bought from Millipore Sigma, Burlington, MA, USA; and sequencing-grade modified trypsin (cat # V5111a) was purchased from Promega Corporation, Madison, WI, USA. All other chemicals used in this study were purchased from Sigma-Aldrich, St. Louis, MO, USA.

### 2.2. Semen Processing, Sperm In Vitro Capacitation (IVC), and Mouse Tissue Collection

Fresh boar spermatozoa were collected weekly from healthy, non-transgenic fertile large white boars (*n* = 3) at the National Swine Resource and Research Center, University of Missouri, which is used for routine in vitro fertilization trials with high blastocyst yield. The sperm-rich fraction (*n* = 6) was used for the study purposes. The concentration and motility of ejaculates were evaluated using conventional semen analysis methods under a light microscope. The sperm concentration was measured using a hemocytometer (ThermoFisher Scientific) and ranged from 300 to 500 million/mL; only ejaculates with >80% motile spermatozoa and <20% morphological abnormalities were used for the study. Collections contaminated with urine were discarded. The fractions were free of contaminants other than the expected minimal content of sperm cytoplasmic droplets, thus not necessitating gradient purification. Spermatozoa were separated from seminal plasma using centrifugation (2000 RPM ~ 400× *g*, 10 min; IEC Centra CL2, ThermoFisher Scientific). Spermatozoa were washed with pre-warmed HEPES buffered Tyrode lactate medium supplemented with polyvinyl alcohol (TL-HEPES-PVA) containing 10 mM sodium lactate, 0.2 mM sodium pyruvate, 2 mM NaHCO_3_, 2 mM CaCl_2_, 0.5 mM MgCl_2_, and 0.01% (*w*/*v*) polyvinyl alcohol (PVA); pH = 7.4, t = 37 °C. After the final wash, the sperm pellet was split into quarters; the first quarter was not allowed to capacitate, directly processed as described below, and used as a non-capacitated (NC) control sample. The other three quarters were in vitro capacitated (IVC, 4 h, 37 °C, 5% CO2) in the modified HEPES-buffered medium described above, supplemented with 5 mM sodium pyruvate, 11 mM glucose, and 2% (*w*/*w*) BSA, as described previously [50], either in the absence of the proteasomal inhibitor (IVC control), in the presence of 100 µM MG132 and 0.2% ethanol (IVC, proteasomally inhibited group), or in the presence of 0.2% ethanol alone (IVC vehicle control). Spermatozoa after IVC were washed to remove the BSA. Non-capacitated and IVC-capacitated groups were processed for (i) mass spectrometry of the sperm surface proteins, (ii) immunocytochemistry and image-based flow cytometry, and (iii) Western blotting.

Mouse kidney, mouse liver, and mouse bladder tissues were kindly donated by Dr. Lei Lei, Department of Obstetrics, Gynecology and Women’s Health, University of Missouri, Columbia, MO 65211, USA.

### 2.3. Biotin-Directed Affinity Purification of Sperm Surface Protein

The isolation of sperm surface proteins from non-capacitated, IVC, proteasomally inhibited IVC, and vehicle control IVC spermatozoa was done as we described earlier [53]. Briefly, a Thermo Scientific Pierce Cell Surface Protein Isolation kit was used according to the manufacturer’s protocol. In this method, mammalian cells were first labeled with EZ-Link Sulfo-NHS-SS-Biotin, which is a thiol-cleavable amine-reactive biotinylation reagent. Cells were subsequently lysed with a mild detergent, and labeled proteins were then isolated with NeutrAvidin^TM^ immobilized on agarose beads. The bound proteins were released via incubation with an SDS loading buffer (50 mM Tris∙HCl, pH 6.8, 1% (*v*/*v*) glycerol, 2% (*w*/*v*) SDS) containing 100 mM DTT. To achieve satisfactory protein yields, three column equivalents per treatment group were used for a single MS run. Three volumes of acetone were added to each extract and the samples were stored at −25 °C prior to the high-resolution mass spectrometry.

### 2.4. Proteomics

Sperm surface, acetone-precipitated protein pellets from all four treatment groups were washed with 80% cold acetone twice. The pellets were dissolved in 10 µL of urea buffer (6 M urea, 2 M thiourea, and 100 mM ammonium bicarbonate). The solubilized protein was reduced using DTT and alkylated using iodoacetamide. Next, trypsin (cat # V5111a, Promega) was added for overnight digestion at 4 °C. The digested peptides were desalted using ZipTip with C18 resin (cat # ZTC18M096, Millipore Sigma), lyophilized, and resuspended in 10 µL of 5/0.1% acetonitrile/formic acid. One microliter of the peptide solution was loaded on a C18 column (20 cm × 75 µm, 1.7 µm) of nanoElute LC-MS system in tandem connection with a timsTOF Pro MS/MS system (LC-MS + MS/MS, Bruker Co., Billerica, MA, USA) with a step gradient of acetonitrile at 300 nL/min. The following LC gradient conditions were applied: The initial conditions were 2% B (and 98% A) (A: 0.1% formic acid in the water; B: 99.9% acetonitrile, 0.1% formic acid), followed by a 20 min ramp to 17% B (and 83% A), 17–25% B (and 83-75% A, respectively) over 27 min, 25–37% B (and 75-63% A, respectively) over 11 min, a gradient of 37% B (and 63% A) to 80% B (and 20% A) over 6 min, and hold at 80% B (and 20% A) for 6 min. The total duration of the LC run time was 70 min, and the MS data were collected over an m/z range of 100 to 1700. During the MS/MS data collection, each TIMS cycle included 1 MS + an average of 10 PASEF MS/MS scans. Raw data were searched using PEAKS (version X) with the UniProtKB Sus scrofa protein database downloaded on Mar 01, 2019, with 88,374 entries. The data search was adjusted for trypsin digestion with two initial missed cleavage sites, fixed modification by carbamidomethylation, and variable modification by methionine oxidation and pyroglutamate formation at peptide N-terminal glutamic acid. The precursor ion mass error tolerance was ±20 ppm and the MS/MS fragment ion mass error tolerance was ±0.1 Da. For the identification of proteins, the following criteria were used: ≥1 unique peptide and ≥2 peptides per protein; the false discovery rate was set to 1%. The proteomic study was done in one pilot run and three additional full replicates. Each run used a new sperm collection. Searched protein and peptide result files from all four runs were exported from PEAKS to spreadsheets and are available upon request. Identified proteins with measured spectral counts were combined, and the protein spectrum number was normalized based on the total spectrum number and internal control protein (sperm head acrosin) in each sample. The pilot replicate was excluded from the data analysis due to the unsatisfactory number of total proteins identified, which was the result of a relatively short LC run (30 min), as opposed to the 70 min used in the three working replicates. The data from three working replicates were used for data analysis. The results are presented in the Excel spreadsheet as Appendix A.

### 2.5. Immunofluorescence

Non-capacitated and IVC-capacitated spermatozoa with and without proteasomal inhibition were fixed in 2.0% (*v*/*v*) formaldehyde in PBS for 10 min at room temperature, and permeabilized in 0.1% TrX-100 in PBS (PBST). Approximately 20 million spermatozoa were blocked with PBST supplemented with 5% normal goat serum (NGS). All the antibodies used for immunofluorescence studies were commercial and previously characterized. Primary antibodies used were as follows: anti-CSE1L (1:100 dilution, PA5-86130, Invitrogen), anti-PFDN4 (1:100 dilution, PA5-110100, Invitrogen), and anti-STK39 (1:33 dilution, PA5-17338, Invitrogen), which were all diluted in PBST supplemented with 5% NGS. Primary antibodies were added to sperm sample tubes and incubated overnight at 4 °C. For the primary control, non-immune rabbit serum of comparable immunoglobulin concentration was used instead of primary antibodies and processed in the same fashion. The following morning, spermatozoa were washed twice with PBST with 1% NGS, and an appropriate species-specific secondary antibody, namely, goat anti-rabbit conjugated to AF488 or AF647 (1:200 dilution), in PBST with 1% NGS was added and allowed to incubate for 40 min at room temperature. For acrosomal integrity assessment, peanut agglutinin conjugated to Alexa Fluor 568 (PNA-AF568, 1:1500 dilution; Invitrogen, Waltham, MA, USA) was used, and 4′,6-Diamidino-2-Phenylindole Dilactate (DAPI), which is a DNA stain (1:1000 dilution), was used as a reference and nuclear contrast stain. Both PNA-AF568 and DAPI were mixed and coincubated with secondary antibodies. After incubation with secondary antibodies, spermatozoa were washed twice with PBST with 1% NGS. For the secondary antibody control, secondary antibodies were omitted, as well as DAPI and PNA-AF647, and processed in the same fashion. All steps were accomplished in suspensions.

### 2.6. Epifluorescence Microscopy

The fluorescently labeled samples were mounted on microscope slides with VectaShield (Vector Laboratories, Inc., Burlingame, CA, #H-1000) and imaged using a Nikon Eclipse 800 microscope (Nikon Instruments, Inc., Melville, NY, USA) with Retiga QI-R6 camera (Teledyne QImaging, Surrey, BC, Canada) operated with MetaMorph 7.10.2.240. software (Molecular Devices, San Jose, CA, USA). Images were adjusted for contrast and brightness in Adobe Photoshop 2020 (Adobe Systems, Mountain View, CA, USA) to match the fluorescence intensities viewed through the microscope eyepieces.

### 2.7. Image-Based Flow Cytometry (IBFC)

The fluorescently labeled samples were measured with an Amnis FlowSight Imaging Flow Cytometer (AMNIS Cytek Biosciences, Fremont, CA, USA), as described previously [54]. The instrument was fitted with a 20x microscope objective (numerical aperture of 0.9) with an imaging rate of up to 2000 events per second. The sheath fluid was PBS (without Ca^2+^ or Mg^2+^). The flow-core diameter and speed were 10 µm and 66 mm per second, respectively. The raw image data were acquired using INSPIRE^®^ software (AMNIS Cytek Biosciences, Fremont, CA, USA). To produce the highest resolution, the camera setting was at 1.0 µm per pixel of the charge-coupled device. Samples were analyzed by using four lasers concomitantly: a 405 nm line (20 mW), a 488/642 nm line (20 mW), a 561 nm line (20 mW), a 785 nm line (70 mW, side scatter), as well as two LEDs (32.57 mW and 19.30 mW). Signals were observed in the following channels: channels 1 and 9—brightfield, channel 2—green fluorescence (AF488, 505–560 nm), channel 4—orange fluorescence (AF568, 595–642 nm), channel 6 (SSC), channel 7—blue fluorescence (DAPI, 435–505 nm), and channel 11—infrared fluorescence (AF647, 642–745 nm). A total of 10,000 events were collected per sample, and data were analyzed using IDEAS^®^ software (Version 6.2.64.0; AMNIS Cytek Biosciences, Fremont, CA, USA). A focused, single-cell population gate was used for the histogram display of mean pixel intensities by frequency for the collected channels. Intensity histograms of individual channels were then used for drawing regions of subpopulations with varying intensity levels and visual confirmation. The intensity of DAPI was used for histogram normalization among the non-capacitated and IVC spermatozoa. Autofluorescent debris was excluded by applying masks.

### 2.8. Protein Isolation from Spermatozoa and Tissues

The processing of an ejaculate to obtain spermatozoa and sperm IVC was described earlier. A total of 100 million PBS-washed spermatozoa per treatment (non-capacitated, IVC spermatozoa with or without proteasomal inhibition, vehicle control) were resuspended in 100 µL of lithium dodecyl sulfate (LDS) loading buffer (106 mM Tris HCl, 141 mM Tris base, 2% LDS, 10% glycerol, 0.51 mM (0.75%) EDTA, 0.22 mM (0.075%) Coomassie Brilliant Blue G250, 0.175 mM (0.025%) Phenol Red, pH = 8.5) supplemented with protease and phosphatase inhibitor cocktail, and left to incubate overnight at 4 °C with end-over-end rotation. The following day, the suspensions were spun at 13,000× *g* and 4 °C for 15 min. Supernatants were transferred to new Eppendorf tubes and before the PAGE loading, 2-mercaptoethanol was added to the final concentration of 2.5% (*v*/*v*), and the samples were incubated at 70 °C for 10 min.

Mouse tissues (kidney, liver, and bladder; ~50 mg/tissue) were chopped into small pieces and homogenized using a mortar and pestle. Proteins were extracted by adding 2× concentrated LDS loading buffer, supplemented with Halt^TM^ Protease and Phosphatase inhibitor cocktail, and left to incubate overnight at 4 °C with end-over-end rotation. The following day, the suspensions were spun at 13,000× *g* and 4 °C for 15 min. Supernatants were transferred to new Eppendorf tubes, and before the PAGE loading, the solutions were adjusted to 1× LDS buffer with ultrapure water, 2.5% (*v*/*v*) 2-mercaptoethanol was added, and the samples were incubated at 70 °C for 10 min.

### 2.9. Immunoprecipitation

The STK39 protein was immunoprecipitated from non-capacitated and IVC spermatozoa using the Thermo Scientific Pierce crosslink magnetic IP/Co-IP kit (cat # 88805) according to the manufacturer’s protocol. Briefly, 5 µg of mouse monoclonal anti-STK39 antibody (cat # NBP1-48021) was bound to the Protein A/G magnetic beads. Conventional IP was performed by omitting antibody cross-linking.

Sperm proteins were extracted from 100 million NC or IVC spermatozoa by the addition of 500 µL of lysis buffer supplemented with Halt^TM^ Protease and Phosphatase inhibitor cocktail and left to incubate for 30 min at 4 °C with end-over-end rotation. The protein concentration was determined with a BCA assay (cat # 23227, ThermoFisher Scientific). One milligram of the total protein was added to the magnetic beads with the bound antibody and incubated for 60 min at room temperature with end-over-end rotation. The antigen was eluted with a low pH and neutralized. Resultant immunoprecipitates were concentrated using MilliporeSigma Amicon centrifugal filters (cat # UFC500396). A 4× LDS sample buffer and 2-mercaptoethanol were added to the immunoprecipitates to prepare a 1× LDS sample buffer and 2.5% (*v*/*v*) 2-mercaptoethanol, respectively. Samples were incubated at 70 °C for 10 min prior to WB detection.

### 2.10. Western Blotting

For polyacrylamide gel electrophoresis (PAGE), a NuPAGE^®^ electrophoresis system was used (Invitrogen, Carlsbad, CA). A protein equivalent of 10 million spermatozoa (~20 µg of the total protein) and ~20 µg of the total protein extracted from mouse tissues was loaded per lane. The PAGE was carried on NuPAGE^TM^ 4–12% Bis-Tris gel (cat# NP0329BOX, Invitrogen, Carlsbad, CA, USA) using TRIS-MOPS SDS Running Buffer (50 mM Tris Base, 50 mM 3-(N-morpholino)propanesulfonic acid (MOPS), 0.1% SDS, 1 mM EDTA, pH = 7.7). An anode buffer was supplemented with 5 mM NaHSO_3_ to prevent reoxidizing of disulfide bonds. The molecular masses of separated proteins were estimated using Novex^®^ Sharp Pre-stained Protein Standard (cat # LC5800, Invitrogen, Carlsbad, CA, USA) run in parallel. The PAGE was carried out for 5 min at 80 V to let the samples delve into the gel and then for another 60–70 min at 160 V. The power was limited to 20 W. After PAGE, proteins were electrotransferred onto a PVDF Immobilon Transfer Membrane (Millipore Sigma) by using an Owl wet transfer system (ThermoFisher Scientific) at 300 mA for 90 min for immunodetection by using a Bis-Tris-Bicine transfer buffer (25 mM Bis-Tris base, 25 mM Bicine, 1 mM EDTA, pH = 7.2) supplemented with 10% (*v*/*v*) methanol and 2.5 mM NaHSO_3_.

The PVDF membrane with the transferred proteins was blocked with 10% (*w*/*v*) non-fat milk (NFM) in TBS with 0.05% (*v*/*v*) Tween 20 (TBST) and incubated with the primary antibody overnight. Primary antibodies were diluted as follows: anti-CSE1L (1:4000 dilution; PA5-86130, Invitrogen), anti-PFDN4 (1:2000 dilution, PA5-110100, Invitrogen), and anti-STK39 (1:1000 dilution), which were all diluted in TBST supplemented with 5% NFM. The next day, the membrane was incubated for 40 min with an appropriate species-specific secondary antibody, such as the HRP-conjugated goat anti-rabbit antibody (GAR-IgG-HRP, 1:10,000 dilution; cat # 31460 Invitrogen). The membrane was reacted with a chemiluminescent substrate (Luminata Crescendo Western HRP Substrate; Millipore Sigma), and the blot was screened with ChemiDoc Touch Imaging System (Bio-Rad, Hercules, CA, USA) to record the protein bands. Membranes were stripped with Restore WB stripping buffer (cat # 21059) and reprobed with anti-TUBB (clone E7, AB_2315513) for protein normalization purposes. The images were analyzed using Image Lab Touch Software (Bio-Rad, Hercules, CA, USA). Unless specified otherwise, procedures were carried out at room temperature. Membranes were stained with CBB R-250 after chemiluminescence detection for additional protein load control.

### 2.11. Statistical Analysis

Four replicates (one pilot trial and three working replicates) were conducted for high-resolution mass spectrometry and the data from three working replicates were used for MS data analysis. Four IBFC replicates and six WB replicates were performed to ensure scientific rigor. The numbers of replicates are also denoted in the figure legends. Each data point is presented as a mean ± SD. Datasets were tested for normal distribution by using the Shapiro–Wilk normality test and processed using one-way analysis of variance (ANOVA) with GraphPad Prism 8.02 (GraphPad Software, Inc., La Jolla, CA, USA) in a completely randomized design. Sidak’s multiple comparison test was used to compare the mean values of individual treatments with a 95% confidence interval. A value of *p* < 0.05 was considered statistically significant.

## 3. Results

### 3.1. Identified Sperm Surface Proteins of Non-Capacitated and Capacitated Spermatozoa

A fresh, sperm-rich fraction of ejaculates (*n* = 6) from fertile, non-transgenic white large boars (S. scrofa, *n* = 3) was used to conduct this experiment. The sperm-rich fraction was split into quarters, where three quarters were in vitro capacitated (IVC) under (i) proteasomal-permissive conditions, (ii) 100 µM MG132 proteasomal-inhibiting conditions including 0.2% (*v*/*v*) EtOH, and (iii) 0.2% EtOH vehicle control, while the last quarter was used as a non-capacitated (NC) control.

Using the biotin-directed affinity purification approach, sperm surface subproteomes from all three IVC sperm groups and NC spermatozoa were extracted with a mild detergent and isolated using a NeutrAvidin^TM^ agarose column. Four biological replicates of sperm IVC and surface biotinylation/pulldowns were performed, and proteomic analysis was performed in four biological replicates, including one pilot and three working replicates that were used for statistical and bioinformatic analyses. Altogether, we identified 3779 UniProtKB-annotated accessions (Appendix A), which translated into 1991 gene-encoded proteins (1680 VGNC annotated gene products). Further, 1209 UniProtKB-annotated accessions (1072 VGNC annotated gene products) that contained 3-mercaptopropionate residues (covalent tag after sperm surface biotinylation) were considered for further analysis.

The identified sperm surface protein dataset was assessed via GO enrichment analysis, including biological process, molecular function, and cellular component analyses, as well as the KEGG pathway to classify the identified proteins. The GO biological process enrichment analysis (Figure 1 and Appendix A) identified 112 categories, with some of the most enriched ones being *protein folding* (31 protein hits, *p* = 2.66 × 10^−15^), *binding of sperm to zona pellucida* (25 protein hits, *p* = 4.4 × 10^−22^), and *flagellated sperm motility* (22 protein hits, *p* = 8.05 × 10^−13^). The GO cellular component enrichment analysis (Figure 2 and Appendix A) identified 86 different categories, with some of the most enriched categories being *cytoplasm* (245 protein hits, *p* = 5.1 × 10^−14^), *mitochondrion* (156 protein hits, *p* = 2.16 × 10^−52^), and *membrane* (62 protein hits, *p* = 4.29 × 10^−3^). Furthermore, the GO molecular function enrichment analysis (Figure 3 and Appendix A) identified 96 different categories, with some of the most enriched categories being *ATP binding* (123 protein hits, *p* = 5.7 × 10^−12^), *ATPase activity* (36 protein hits, *p* = 1.57 × 10^−7^), and *unfolded protein binding* (31 protein hits, *p* = 1.39 × 10^−18^). Finally, the KEGG pathway analysis (Figure 4 and Appendix A) identified 86 enriched pathways, with some of the most represented categories being *metabolic pathways* (194 protein hits, *p* = 1.68 × 10^−35^); *multiple pathways of neurodegeneration* (105 protein hits, *p* = 2.39 × 10^−39^), including Amyotrophic lateral sclerosis, Huntington’s disease, Parkinson’s disease, Alzheimer’s disease, prion disease, and spinocerebellar ataxia pathways); and *proteasome pathway* (31 protein hits, *p* = 1.1 × 10^−27^).

### 3.2. Changes in Sperm Surface Subproteome after In Vitro Capacitation (Non-Capacitated vs. Capacitated Sperm Surface Subproteomes)

Changes in sperm surface protein abundances of the IVC spermatozoa were investigated by using volcano plot analysis [55] and the results are presented in Figure 5a. Student’s *t*-test was used to identify additional different sperm surface proteins between the NC and IVC sperm surface subproteome. The fold change threshold was set to 1.5, and the level of significance was *p* < 0.05. After the spermatozoa were capacitated, 141 proteins were more abundant, while 91 proteins were less abundant on the surface of the IVC spermatozoa when compared with NC spermatozoa. These significantly different proteins were subjected to GO enrichment analysis (Figure 6, Figure 7 and Figure 8). Some of the most enriched biological processes that were on the surface of the capacitated spermatozoa were *protein folding* (seven protein hits, *p* = 5.0 × 10^−5^), *tricarboxylic acid cycle* (four protein hits, *p* = 8.4 × 10^−4^), and *malate metabolic process* (three protein hits, *p* = 1.3 × 10^−3^). Meanwhile, the *proteasome-mediated ubiquitin-dependent protein catabolic process* (five protein hits, *p* = 2.7 × 10^−3^), *protein targeting to nuclear inner membrane* (two protein hits, *p* = 8.7 × 10^−3^), and *protein transport* (five protein hits, *p* = 1.4 × 10^−2^) processes were some of the most downregulated ones. Some of the most enriched cellular compartments harboring proteins that were on the surface of the IVC spermatozoa were *mitochondrion* (19 protein hits, *p* = 4.1 × 10^−6^), *cytoplasm* (36 protein hits, *p* = 1.2 × 10^−4^), and *motile cilium* (5 protein hits, *p* = 1.3 × 10^−4^), while the downregulated processes were *nuclear pore* (4 protein hits, *p* = 4.4 × 10^−4^), *kinetochore* (4 protein hits, *p* = 4.3 × 10^−3^), and *nuclear membrane* (4 protein hits, *p* = 1.3 × 10^−2^). Molecular functions that were among the most enriched on the surface of the IVC spermatozoa were *unfolded protein binding* (seven protein hits, *p* = 1.9 × 10^−5^), *ubiquitin protein ligase binding* (seven protein hits, *p* = 4.8 × 10^−3^), and *ATP binding* (seven protein hits, *p* = 6.4 × 10^−3^).

### 3.3. Targets of the Ubiquitin–Proteasome System on the Sperm Surface during In Vitro Capacitation (Sperm Surface Subproteomes of Capacitated, Proteasomally Inhibited vs. Capacitated, Vehicle Control Spermatozoa)

In line with the previous section, changes in sperm surface protein abundances between IVC proteasomally inhibited and IVC vehicle control spermatozoa were compared using volcano plot analysis (Figure 5b). Student’s *t*-test was used to identify additional different sperm surface proteins. The fold change threshold was set to 1.5, and the level of significance was *p* < 0.05. In total, 14 proteins were found to be significantly different between the proteasomally inhibited and vehicle groups. Seven proteins were more abundant on the surface of the proteasomally inhibited spermatozoa, including NIF3L1, CSE1L, NDUFB7, PGLS, PPP4C, STK39, and TPRG1L. The more abundant proteins on the vehicle control spermatozoa were BPHL, GSN, GSPT1, PFDN4, STYXL1, TIMM10, and UBXN4 (Table 1).

A literature search of these 14 proteins was performed by searching each protein individually in the PubMed.gov database for their reported/known functions and localizations, specifically in spermatozoa (Table 1). The GSN and NDUFB7 proteins (14.3%) have known biological functions in spermatozoa; the BPHL, PFDN4, PGLS, PPP4C, STK39, STYXL1, and TIMM10 (50%) functions were assumed based on their somatic counterparts; and CSE1L, GSPT1, NIF3L1, TPRG1L, and UBXN4 (35.7%) had no known function in spermatozoa. Localization-wise, CSE1L, PFDN4, and STK39 (21.4%) are localized in the head; BPHL, NDUFB7, and TIMM10 (21.4%) are localized in the midpiece/mitochondrial sheath; GSN was reported in both the head and flagellum; and finally, GSPT1, NIF3L1, PGLS, PPP4C, STYXL1, TPRG1L, and UBXN4 (50%) have no known localization in spermatozoa. Three significantly different surface proteins (CSE1L, PFDN4, and STK39) between the capacitated proteasomally inhibited and vehicle control groups were phenotyped in spermatozoa within all treatment groups.

### 3.4. Localization and Dynamics of Selected Sperm Surface UPS-Regulated Proteins during In Vitro Sperm Capacitation

Chromosome segregation 1-like protein (CSE1L, also known as Exportin-2, or Importin-alpha re-exporter), prefoldin subunit 4 (PFDN4), and non-specific serine/threonine protein kinase 39 (STK39, EC:2.7.11.1, also known as SPAK), which are the candidates for UPS modulation during sperm in vitro capacitation, were subjected to phenotype studies by the means of immunocytochemistry (ICC), image-based flow cytometry (IBFC), Western blotting (WB), and immunoprecipitation (IP; only STK39) (Figure 9, Figure 10 and Figure 11).

Starting with CSE1L, we observed faint localization in the acrosomal region of both the non-capacitated (Figure 9a) and in vitro capacitated (Figure 9a’) spermatozoa. Furthermore, CSEL1 was also present in the post-acrosomal segment of the NC spermatozoa while absent from the IVC spermatozoa. After the spermatozoa were IVC, a decrease in fluorescence intensity median was observed using IBFC, resulting in two distinct sperm cohorts (Figure 9b). The decrease in the CSE1L signal intensity was observed in the acrosomal area, as well as the post-equatorial segment (Figure 9b’,b”), which confirmed the ICC images. The CSEL1 sperm cohort redistribution was significantly reduced (*p* = 0.5, Figure 9b,b’’’) when the spermatozoa were capacitated under proteasome-inhibiting conditions (100 µM MG132), compared with the vehicle control group (0.2% EtOH). By using WB, we detected CSE1L in NC spermatozoa as a doublet at ~56 kDa and 60 kDa (Figure 9c–c”). The CSEL1 abundance decreased significantly after IVC (*p* = 0.02, Figure 9c’’’), while WB detected no significant abundance change between the proteasomally inhibited IVC group vs. the vehicle control group (*p* = 0.94, Figure 9c’’’).

The second studied protein, namely, PFDN4, was localized to the apical ridge region of the sperm head acrosome in NC spermatozoa (Figure 10a), and the signal intensity in the same area was amplified in the IVC spermatozoa (Figure 10a’). The median signal intensity was increased after IVC, resulting in two distinct sperm cohorts (Figure 10b). The higher median signal intensity was attributed to the stronger fluorescence intensity in the apical ridge region (Figure 10b’ vs. Figure 10b”), as was also observed using ICC. The sperm cohort redistribution was significantly hindered (*p* = 0.05) in the proteasomally inhibited IVC treatment group compared with its vehicle IVC control (Figure 10b,b’’’). We detected multiple bands by using WB. By using mouse liver and kidney tissue controls, we mapped the PDNF4 to the molecular weight of ~17 kDA (Figure 10c,c’,c”). The abundance of the 17 kDa band was reduced in a non-significant manner after IVC (*p* = 0.31), and proteasomal inhibition had no significant effect on the 17 kDa band density when compared with the vehicle control (*p* = 0.39, Figure 10c’’’).

The last studied protein, namely, STK39, was below the detection threshold in NC spermatozoa (Figure 11a), while a faint signal in the acrosomal region was observed in IVC spermatozoa by using ICC (Figure 11a’). The increase in the fluorescence signal intensity in the acrosomal region after IVC was paralleled by a significant increase in the median fluorescence intensity (*p* = 0.002, Figure 11b,b’’’). The acrosomal labeling of STK39 was detected by using IBFC (Figure 11b’) and confirmed in IVC spermatozoa (Figure 11b’’’). The median fluorescence intensity increase was decelerated in the proteasomally inhibited IVC group (*p* = 0.09) compared with its vehicle control IVC group (Figure 11b,b’’’). The STK39 protein was below the detection limits of WB in the sperm extracts. Therefore, STK39 was immunoprecipitated from non-capacitated and in vitro capacitated spermatozoa. By using WB, STK39 was detected at ~74 kDa in immunoprecipitates originating from both NC and IVC spermatozoa. Additionally, a 43 kDa band was detected in the immunoprecipitate originating from IVC spermatozoa.

## 4. Discussion

The present study monitored the sperm surface subproteome changes during boar sperm capacitation, with a particular focus on the substrates of sperm proteasomes. We used our established in vitro capacitation protocol [77] to capacitate pig spermatozoa under proteasome-permissive and -inhibiting conditions, including non-capacitated and vehicle controls. In order to selectively isolate sperm surface proteins, we used our established sperm biotin-mediated affinity purification protocol [53,78], which was originally developed by Zhao et al. [79] to study integral plasma membrane proteins. Unlike Zhao et al., we were interested in plasma membrane-associated proteins as well, and therefore, we omitted the third step when such proteins are depleted by harsh washes with high-salt and high-pH buffers [79]. This approach was used by another research group to map bovine sperm surface proteins [80]. By using high-resolution mass spectrometry, we identified 1680 VGNC annotated proteins that were associated with the sperm surface. To ascertain that these proteins came exclusively from the sperm surface, or the outermost layer of the sperm coating proteins that were accessible to Sulfo-NHS-SS-Biotin, 1072 proteins that had a residual tag after biotinylation (3-mercaptopropionate) were considered for further analysis. The reason for the exclusion of 608 proteins was to eliminate proteins that might have interacted with biotinylated proteins captured to the streptavidin column during affinity purification. We are aware that peptides void of 3-mercaptoproionate that were obtained after chymotrypsin digestion might have been derived from proteins that were surface biotinylated. In other words, to minimize false positive surface proteins, we might have excluded false negative surface proteins.

To characterize the global sperm surface proteome, we used GO enrichment analysis, including biological processes, cellular components, and molecular functions, as well as KEGG pathway analysis. We found 116 enriched GO biological process terms (*p* < 0.05) with those previously annotated in sperm biology, including the binding of sperm to zona pellucida (the most enriched process, validating our experimental design), multiple metabolic processes (including glycolysis and pyruvate metabolism), sperm flagellar motility, the proteasomal protein-catabolic process (both ubiquitin-dependent and -independent), fertilization, sperm capacitation, the fusion of sperm to the egg plasma membrane, acrosome reaction, and response to the hormone. A total of 89 GO cellular component terms were enriched (*p* < 0.05), with the three most enriched components being mitochondrion, sperm flagellum, and acrosomal vesicle. It might come as a surprise that the sperm surface is the compartment that is most enriched with mitochondrion component proteins; however, this exposure of the sperm mitochondrial sheath might be a preparative step for post-fertilization sperm mitophagy [81]. In our most recent study, we showed that sperm surface proteins, such as FUNDC2 and BAG5, accumulate on the sperm mitochondria [82]. Equally, sperm surface proteins are enriched with acrosomal vesicle components as well. Similar to mitochondrial components, it was shown previously that acrosomal proteins, such as ZAN, ACR, ACRBP, ZPBP1, and ZP3R, traffic to the sperm head surface during sperm capacitation [83,84,85]. Further to the aforementioned 89 GO cellular component terms, 105 GO molecular function terms were enriched (*p* < 0.05) with a myriad of binding functions, structural functions, and enzymatic activities. Lastly, we performed a KEGG pathway analysis to identify what pathways might be active on the sperm surface, yielding a total of 89 surface-enriched pathways (*p* < 0.05). The most enriched were multiple pathways of neurodegeneration, including Alzheimer’s, Huntington’s, and Parkinson’s disease pathways. Our and other groups already reported the involvement of neurodegenerative disease-associated pathways in sperm physiology [77,86], which reflects similar gene expression between spermatozoa and neurons [87,88]. Other pathways that were enriched on the sperm surface were multiple metabolic, proteasomal, and signaling pathways, including HIF-1, PPAR, and cGMP-PKG.

To identify the differences in sperm surface subproteomes between NC and IVC spermatozoa, we employed volcano plot analysis. We found 232 significantly different proteins (*p* < 0.05) with at least a 1.5-fold difference. More specifically, 141 proteins were more abundant, while 91 proteins were less abundant on the surface of IVC spermatozoa. We performed the same GO enrichment analyses to identify which biological process, cellular component, and molecular function terms were enriched in the sperm surface proteome of NC and IVC spermatozoa. The biological process terms that were enriched (*p* < 0.05) in IVC spermatozoa included protein folding (these included chaperone and chaperonin proteins that are components of multiprotein zona-binding complexes [89,90,91]), multiple metabolic processes, fertilization, and acrosomal vesicle exocytosis. These results confirmed that the sperm surface proteome is remodeled in preparation for the oocyte encounter [84,92]. Biological process terms that were enriched (*p* < 0.05) in NC spermatozoa included proteasome-mediated ubiquitin-dependent proteolytic catabolism and several processes including the nucleus, which was due to the fact that several nucleoporins were detected that might have a different role in spermatozoa that in somatic cells. For instance, the sperm nuclear envelope is mostly devoid of nuclear pore complexes, but some are observed in the redundant nuclear envelopes surrounding the sperm tail connecting piece [93]. We previously reported that the abundance of proteasomes decreases after capacitation [50], suggesting that proteasomes are important in the early stages of capacitation. Cellular component terms that we found to be enriched (*p* < 0.05) in IVC spermatozoa were mitochondrion (presumably a preparative step for post-fertilization sperm mitophagy, as discussed above), cytosol/cytoplasm (these proteins might have a different localization in spermatozoa, for instance, RAB2A, which is a cytoplasmic protein in a somatic cell and is localized on the sperm surface [94]), and sperm flagellum. Cellular component terms that we found to be enriched (*p* < 0.05) in NC spermatozoa were nuclear pore/membrane/periphery (with different nucleoporins detected) and cytosol. The function of the sperm-surface-expressed nucleoporins needs to be addressed by future studies, although the redundant nuclear envelopes mentioned above are a likely culprit. Lastly, molecular function terms that we found to be enriched (*p* < 0.05) in IVC spermatozoa were reflective of the enriched biological process terms, including unfolded protein binding, ubiquitin protein ligase binding, ATP binding, and other types of binding. Molecular function terms that we found to be enriched (*p* < 0.05) in NC spermatozoa were a structural constituent of the nuclear pore and GTPase activity/binding.

To search for the sperm surface targets of the ubiquitin–proteasome system (UPS) during sperm IVC, we capacitated boar spermatozoa under proteasomally inhibiting conditions (100 µM MG132) and compared their sperm surface proteome to the one isolated from the IVC vehicle control (0.2% EtOH) spermatozoa. We confirmed the presence of proteasomal subunits on the sperm surface, including PSMA1-8, PSMB1-7, PSMC1-6, PSMD1-9, PSMD11-14, PSME4, and PSMF1. Using volcano plot analysis in combination with Student’s *t*-test (a 1.5-fold change and *p* < 0.05 were the cutoffs), we were able to identify 14 significantly different proteins with more than a 1.5-fold difference. Of these, it seems that the UPS had rather a narrow range of targets on the sperm surface during sperm capacitation. An alternative explanation was offered by a computational study done by Tarashi et al. [49], where the authors suggested sperm capacitation to be a “two-player game”, with the two players being the 26S proteasome and protein kinase A (PKA, PRKA). It is thus possible that under proteasome-inhibiting conditions, sperm capacitation might be driven solely by PRKA activity at a slower pace when compared with the proteasome-permissive vehicle control. This might explain why we were not able to prevent complete sperm capacitation by solely inhibiting the UPS/proteasomal core [22]. We consider it important to mention that MG132 is a potent inhibitor of chymotrypsin-like activity of the 20S core (K_i_ = 2–4 nM) [95]; however, it is not a strictly site-specific inhibitor and it also inhibits caspase-like activity (K_i_ = 900 nM) and trypsin-like 20S core activity (K_i_ = 2700 nM), albeit with lesser efficiency. Even though we used a relatively high concentration of MG132 (100 µM), the residual caspase-like and trypsin-like proteasomal activities might compensate for inhibited chymotrypsin-like activity. Further, as the abundance of the proteasomes drops after capacitation (shown in this study, as well as our previous study [50]), later stages of sperm capacitation might be less dependent on proteasomal activity. These hypotheses require further investigation. Out of 14 significantly different sperm surface proteins, NIF3L1, CSE1L, NDUFB7, PGLS, PPP4C, STK39, and TPRG1L were found to be more abundant on the surface of proteasomally inhibited, IVC spermatozoa; meanwhile, the BPHL, GSN, GSPT1, PFDN4, STYXL1, TIMM10, and UBXN4 were more abundant on the surface of IVC vehicle control spermatozoa. Only two proteins, GSN and NDUFB7 have known functions in sperm capacitation, making the other 12 proteins good candidates for further studies.

We phenotyped three candidate proteins of UPS regulation during sperm capacitation, namely, CSE1L, PFDN4, and STK39. We selected these three candidate target proteins to have representatives of both up- and downregulated targets, and their abundance differences were relatively high, which made them easy to capture by our phenotyping pipeline. Very little is known about the role of CSE1L in sperm capacitation. Chromosome segregation 1-like protein, also named cellular apoptosis susceptibility protein (CAS), is highly expressed in most cancer types. The CSE1L is an exosome/microvesicle membrane protein [96] with multiple roles in apoptosis, cell survival, chromosome assembly, nucleocytoplasmic transport, microvesicle formation, and cancer metastasis [97]. In a recent study by Shi et al. [59], the expression and role of CSE1L were investigated during mouse spermatogenesis. Therein, CSE1L was found to be highly expressed on both mRNA and protein levels in spermatogonia, spermatocytes, and round spermatids. Knockdown of CSE1L in spermatogonia and spermatocyte cell lines resulted in a significant reduction of cell proliferation and an increase in apoptosis [59]. A weak CSE1L expression was observed in the mouse epididymal spermatozoa in the acrosomal region [59]. Similarly, we observed a weak CSE1L expression in the acrosomal region of both non-capacitated and capacitated boar spermatozoa. The immunofluorescence signal of CSE1L decreased further during sperm IVC, which was reflected in the decreased abundance of CSE1L observed by both mass spectrometry (MS) and WB. Interestingly, the median fluorescence intensity decrease was less obvious in the IVC spermatozoa (*p* = 0.05, observed with IBFC), suggesting that CSE1L might be a target of the UPS during sperm capacitation. The presence of CSE1L was also reported in human spermatozoa with normal ROS levels [58]. In the study by Sharma, et al. [58], ROS levels were measured with a chemiluminescence assay using luminol, expressed as relative light units (RLUs)/s/×10^6^ spermatozoa, where ROS levels < 20 RLUs/s/×10^6^ spermatozoa were considered normal. Even though CSE1L is a proposed component of cAMP/PKA and Ras/ERK signaling pathways in melanoma cells [97] and CDK signaling pathway in spermatogonia and spermatocytes, the surface presence of CSE1L in pig spermatozoa implies a different, unique function in sperm physiology. Since this protein was prevalent in human spermatozoa with a lower level of ROS [58], it is plausible that this protein is either very sensitive to ROS levels or the increased levels of ROS are a result of the CSE1L absence. It may therefore be plausible that CSE1L abundance is reduced by the UPS during sperm capacitation so that the capacitation driven by increased ROS production may proceed [98]. Further research is to be dedicated to this hypothesis.

The second explored protein, namely, prefoldin subunit 4, is a component of a hexameric molecular chaperone prefoldin [66,67]. The PFND hexameric complex has the appearance of a jellyfish that captures unfolded proteins with its six long tentacle-like coiled-coil domains [66] and presents them to a group II chaperonin (TriC/CCT) [67,99]. The prefoldin-group II chaperonin (PFDN-TRiC) system is therefore thought to be important for cytosolic proteostasis. Like CSE1L, increased expression of PFDN4 was reported in various types of cancerous tumors, such as colorectal cancer [100,101], hepatocellular carcinoma [102], gastric cancer [103], breast cancer [104], or epithelial ovarian cancer [105]. In this regard, CSE1L and PFDN4 could be novel, previously unrecognized cancer-testis antigens, which was partially supported in the case of CSE1L and its role in maintaining cell proliferation and division in seminomas [106]. Unlike in somatic/cancer cells, we localized PFDN4 in the extracellular space of the apical ridge of the sperm head in non-capacitated spermatozoa, which became more available for immunolabelling after sperm IVC capacitation. This increase in the said immuno-availability was hindered in the spermatozoa capacitated under proteasomal inhibition (*p* = 0.05), suggesting that the UPS was directly or indirectly involved in the events resulting in a higher median fluorescence intensity. The identified sperm surface proteome suggested the presence of PFDN complex (PFDN1, PFDN2, PFND4, and PFDN6 present), as well as the TRiC complex (TCP1 present), on the sperm surface. It is therefore reasonable to suspect the presence of the PFDN–TRiC system on the sperm surface. A likely role of such multi-chaperone complexes might be the assembly of multiprotein zona-binding complexes reported on the acrosomal surface of mouse [89], human [91], and porcine [90] spermatozoa. These complexes also contained TCP1. The subtle PFDN4 abundance changes were only detectable by using MS, where the PFDN4 on the sperm surface decreased by a factor of 3.04 (*p* = 0.14) after the IVC; interestingly, this drop was accelerated when the spermatozoa were capacitated under proteasomal inhibition when compared with the IVC vehicle control (1.79-fold decrease, *p* = 0.02). Altogether, it appears that the UPS was involved in the negative regulation of PFDN4 shedding from the sperm surface. We can speculate that the reason might be to provide an extra time window for the proper assembly of multiprotein zona-binding complexes in the apical region of a pig spermatozoon.

Non-specific serine/threonine protein kinase 39 (STK39, EC 2.7.11.1, also known as SPAK) was the last protein that we dedicated closer attention to. Similar to the previous two examined proteins, the function of STK39 is well-known in somatic cells, where this cytosolic kinase, in synergy with oxidative stress-responsive kinase 1 (OXSR1, also known as OSR1), is essential for the regulation of Na^+^-K^+^-2Cl^-^ cotransporters (NKCCs) [107,108] via the WNK-SPAK/OSR1 pathway [109], and thus, sustaining ion and fluid homeostasis in mammalian cells. The STK39 is expressed at high levels in normal human testes and the prostate [72], as well as in several cancer cell lines [72,73,110,111,112,113]. The importance of STK39 in mouse spermatogenesis was demonstrated by Liu et al. [71], where a Sertoli cell-specific *OSR1*^−/−^ and *STK39*^−/−^ double-knockout male mice displayed infertility with increased germ cell apoptosis and defective spermatogenesis, resulting in Sertoli cell-only syndrome. These double-knockout male mice had a significantly decreased abundance of phosphorylated NKCC1 (the ubiquitous form, which is more active than dephosphorylated form), suggesting that STK39, in cooperation with OSR1, regulates spermatogenesis via the activation of NKCC1. No attention was paid to STK39 regarding sperm physiology, and to the best of our knowledge, this is the first time that STK39 has been reported in spermatozoa. We localized STK39 to the acrosomal region of IVC spermatozoa, while before capacitation, the STK39 was below the limit of detection, most likely due to antigen inaccessibility. The assumed accessibility was confirmed by using IBFC when we were able to detect acrosomal labeling in both NC and IVC spermatozoa, with a significant increase in median fluorescence intensity after capacitation (*p* = 0.002). The STK39 intensity increase was hindered (*p* = 0.09) when the spermatozoa were capacitated under proteasomal-inhibiting conditions when compared with the vehicle IVC control. The total sperm amount of STK39 was below the detection limit of WB; however, MS showed that the STK39 abundance decreased by a factor of 3.32 after the spermatozoa were IVC (*p* = 0.17), and by a factor of 2.67 (*p* = 0.04) when spermatozoa were capacitated under proteasomal inhibition vs. the vehicle IVC control. The role of STK39 on the sperm surface can only be extrapolated at this time. The very obvious role that was also documented during mouse spermatogenesis [71] would be the regulation of NKCC via its phosphorylation, which is important for the capacitation-associated increase in protein tyrosine phosphorylation, at least in mice [114]. It may be that STK39 phosphorylates other targets on the sperm surface or even on the oolemma.

A deep understanding of the ubiquitin–proteasome system during sperm capacitation will benefit assisted reproductive technologies/therapies (ART) in animals and humans, respectively. For instance, the addition of proteasomal inhibitors into boar semen extenders could prevent the identified UPS-targeted proteins from capacitation-related modifications, which might ward off premature sperm capacitation, which is the common culprit of the low fertility of boar liquid semen used for artificial insemination in the swine industry. Conversely, including small molecule activators of the proteasomal activity in the in vitro fertilization media may boost sperm fertility during human ART, particularly for artificial insemination and conventional IVF. Intracytoplasmic sperm injection (ICSI) could benefit from controlled, proteasome-optimized sperm capacitation before individual spermatozoa are selected for the procedure to boost the post-ICSI disassembly of accessory sperm structures, the release of sperm-borne oocyte activating factors, and sperm nuclear decondensation. Lastly, the unique cellular surface exposure of sperm proteasomes and their sensitivity to non-permeant proteasomal inhibitors provides opportunity for the development of pharmacological contraceptives for humans.

## 5. Conclusions

This study compared the pig sperm surface subproteome of non-capacitated and in vitro capacitated spermatozoa with or without proteasomal inhibition. We found 232 proteins that were significantly different in their abundance by at least 1.5× (*p* < 0.05). These proteins were associated with sperm capacitation, hyperactivation, acrosomal exocytosis, and fertilization. We found 14 significantly different proteins in IVC spermatozoa under proteasomal inhibition when compared with the vehicle IVC control (1.5-fold abundance threshold, *p* < 0.05). We further phenotyped CSE1L, PFDN4, and STK39 and confirmed their capacitation-related changes, as detected using mass spectrometry.

In summary, we demonstrated that the UPS regulates the capacitation-associated sperm surface remodeling by targeting/regulating at least 14 individual sperm surface proteins. This new knowledge will help us identify new pharmacological mechanisms to positively or negatively modulate sperm fertilizing ability in food animals and humans.

## Figures and Tables

**Figure 1 biomolecules-13-00996-f001:**
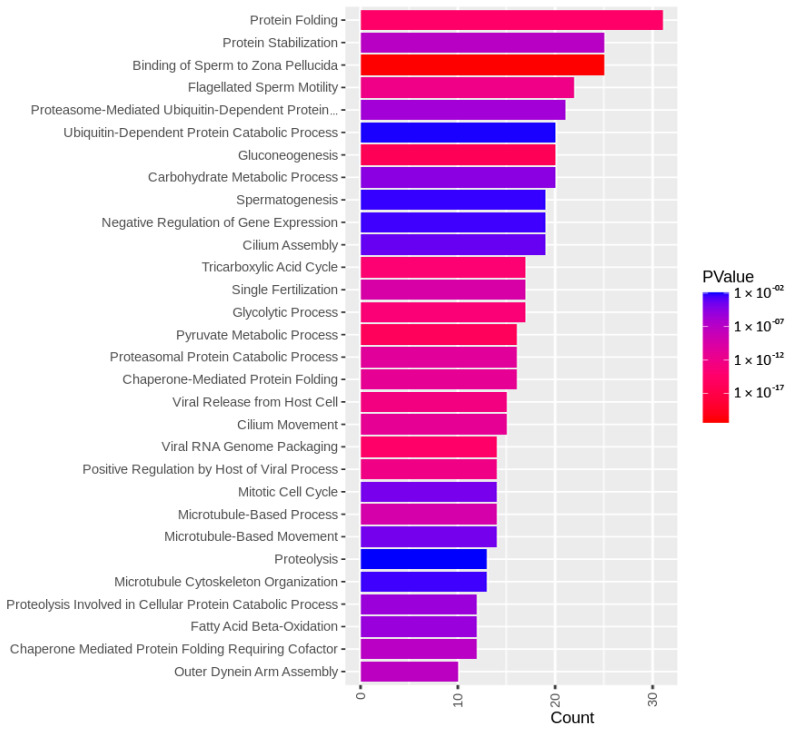
GO biological process enrichment analysis of the identified sperm surface proteins.

**Figure 2 biomolecules-13-00996-f002:**
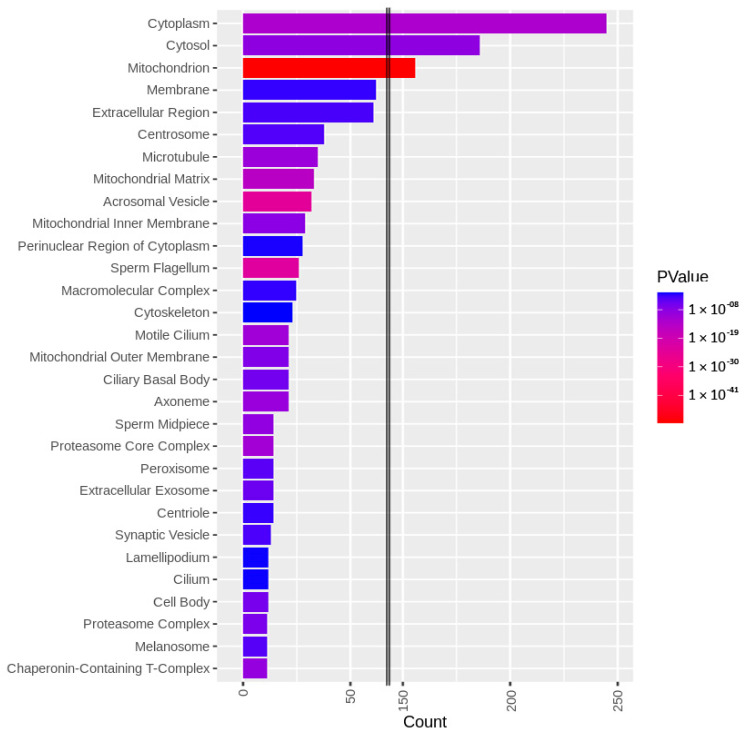
GO cellular component enrichment analysis of the identified sperm surface proteins.

**Figure 3 biomolecules-13-00996-f003:**
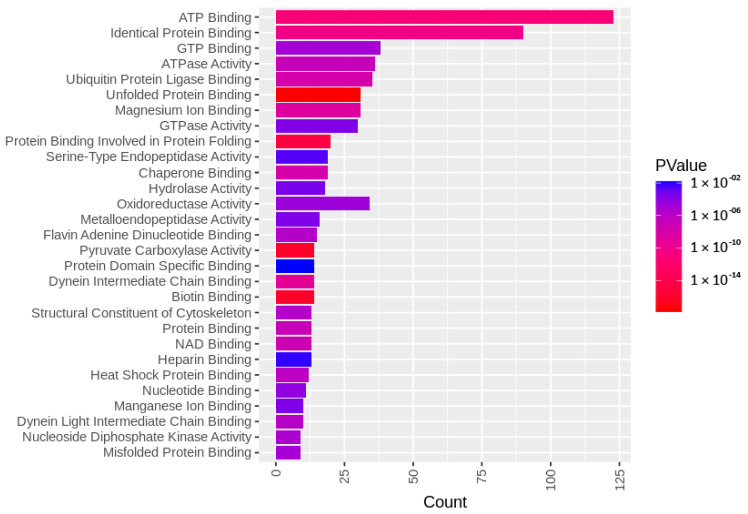
GO molecular function enrichment analysis of the identified sperm surface proteins.

**Figure 4 biomolecules-13-00996-f004:**
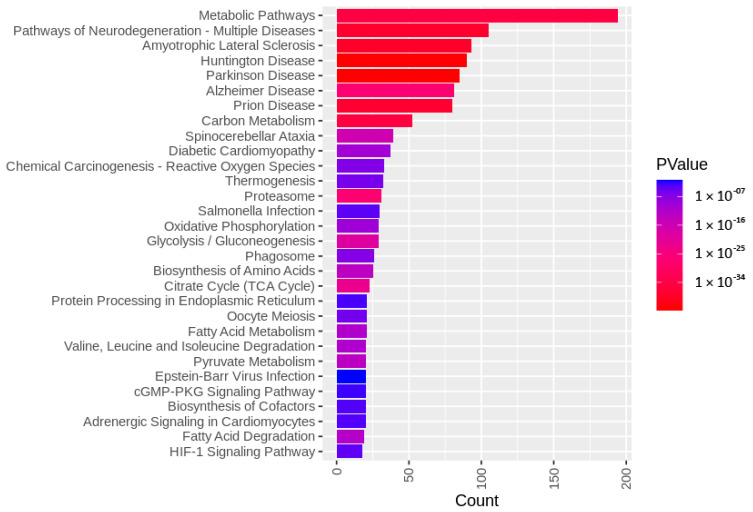
KEGG pathway analysis of the identified sperm surface proteins.

**Figure 5 biomolecules-13-00996-f005:**
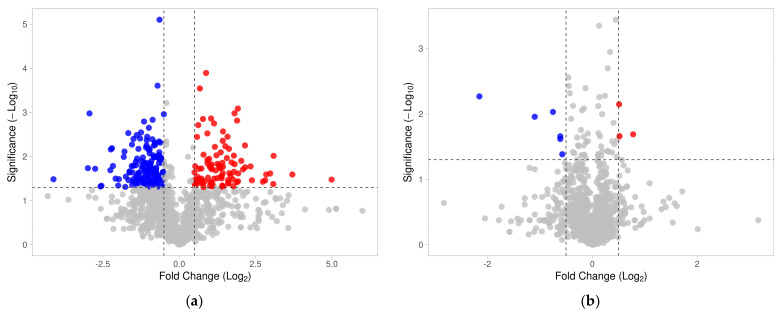
Volcano plot representation of changes in sperm surface protein abundances before and after 4 h of in vitro capacitation (**a**), and IVC spermatozoa with proteasomal inhibition and vehicle control (**b**). Fold change threshold = 1.5 (x-axis) and unadjusted *p*-value ≤ 0.05 (y-axis). Red circles represent sperm surface proteins both above the 1.5-fold change and unadjusted *p*-value threshold; blue circles represent those below the -1.5-fold change and above the unadjusted *p*-value threshold; gray circles represent zincoproteins below the fold change threshold or the unadjusted *p*-value threshold or both thresholds.

**Figure 6 biomolecules-13-00996-f006:**
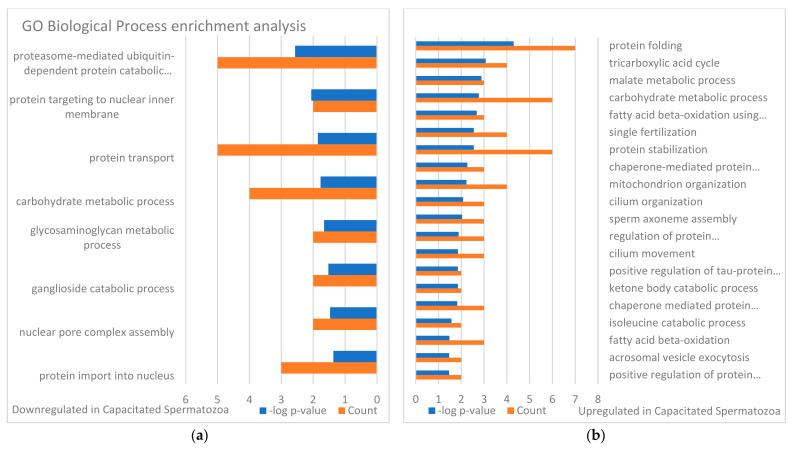
GO biological process enrichment analysis of the identified sperm surface proteins downregulated in the in vitro capacitated spermatozoa (**a**) and upregulated in the capacitated spermatozoa (**b**) when compared with non-capacitated sperm control. The full designation of GO terms that are too lengthy are presented in the Appendix A.

**Figure 7 biomolecules-13-00996-f007:**
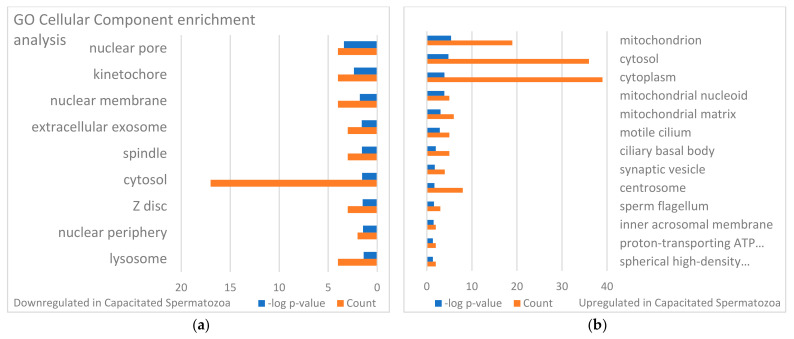
GO cellular component enrichment analysis of identified sperm surface proteins downregulated in the in vitro capacitated spermatozoa (**a**) and upregulated in the capacitated spermatozoa (**b**) when compared with the non-capacitated sperm control. The full designation of GO terms that are too lengthy are presented in the Appendix A.

**Figure 8 biomolecules-13-00996-f008:**
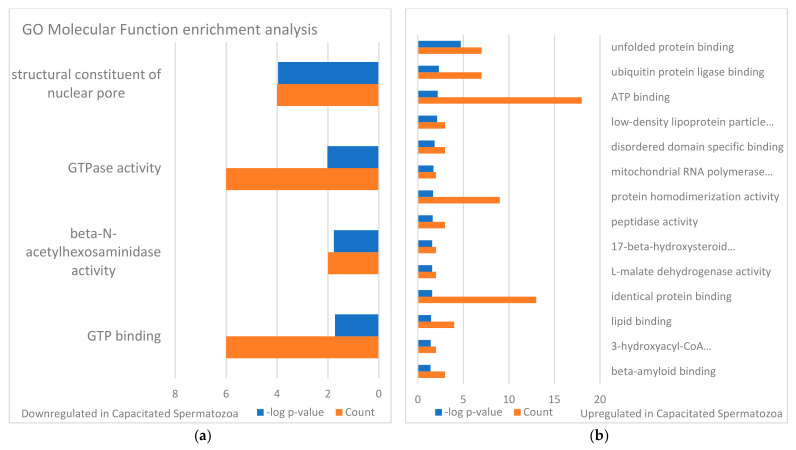
GO molecular function enrichment analysis of identified sperm surface proteins downregulated in the in vitro capacitated spermatozoa (**a**) and upregulated in the capacitated spermatozoa (**b**) when compared with the non-capacitated sperm control. The full designation of GO terms that are too lengthy are presented in the Appendix A.

**Figure 9 biomolecules-13-00996-f009:**
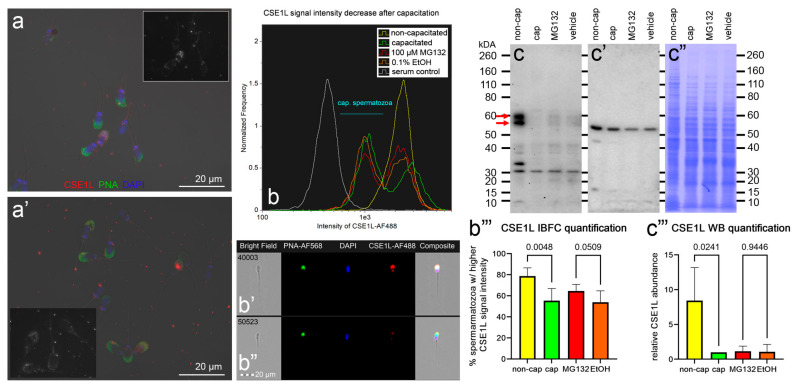
Phenotyping of CSE1L in non-capacitated (NC) and in vitro capacitated (IVC) spermatozoa with and without proteasomal inhibition, including a vehicle control. The ICC detection of CSE1L (red) in NC (**a**) and IVC, non-inhibited (**a’**) spermatozoa; the insets show the CSE1L channel only. Spermatozoa were co-stained for acrosomal integrity with peanut agglutinin lectin (PNA, green) and nuclear stain DAPI (blue). All fluorescence channels were superimposed with the differential interference contrast (DIC) brightfield channel. Scale bars represent 20 µm. The IBFC of formaldehyde fixed and Triton-X-100-permeabilized NC and IVC spermatozoa with or without proteasomal inhibition, including a vehicle and negative, normal rabbit serum (**b**). Immunofluorescence images of NC (**b’**) and IVC spermatozoa (**b”**) obtained by using IBFC to confirm the labeling. The IBFC was performed in 4 replicates with consistent results. Detection of CSE1L using WB (**c**) in the extracts obtained from NC and IVC spermatozoa under proteasome permissive and inhibiting conditions, including a vehicle control. The red arrows point to the CSE1L doublet (predicted molar weight of 64 kDa). The membrane was stripped and reprobed with anti-TUBB antibody (**c’**) and stained with CBB R-250 (**c”**) for protein normalization purposes. Western blotting was performed in six replicates. Statistical evaluation of the IBFC results (**b’’’**) and WB results (**c’’’**) was performed by using ANOVA with Sidak’s post hoc test (α = 0.05).

**Figure 10 biomolecules-13-00996-f010:**
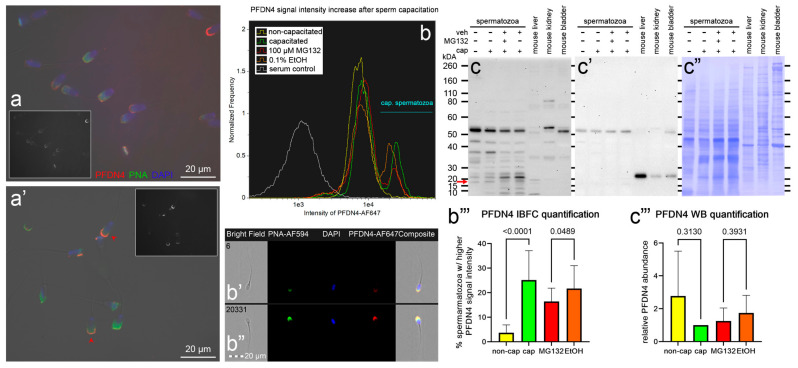
Phenotyping of PFDN4 in non-capacitated (NC) and in vitro capacitated (IVC) spermatozoa with and without proteasomal inhibition, including a vehicle control. The ICC detection of PFDN4 (red) in NC (**a**) and IVC/non-inhibited (**a’**) spermatozoa; the insets show the PFDN4 fluorescence channel only; the red arrowheads indicate capacitated spermatozoa. Spermatozoa were co-stained for acrosomal integrity with peanut agglutinin (PNA lectin, green) and nuclear stain DAPI (blue). All fluorescence channels were superimposed with the DIC brightfield channel. Scale bars represent 20 µm. The IBFC of formaldehyde fixed and Triton-X-100-permeabilized NC and IVC spermatozoa with or without proteasomal inhibition, including a vehicle and negative, normal rabbit serum immunolabeling control (**b**). Immunofluorescence images of NC (**b’**) and IVC spermatozoa (**b”**) obtained by using IBFC to confirm the labeling. The IBFC was performed in 4 replicates with consistent results. Detection of PFDN4 using WB (**c**) in the extracts obtained from NC and IVC spermatozoa under proteasome permissive and inhibiting conditions, including a vehicle control and positive tissue control proteins extracted from mouse liver, kidney, and bladder. The red arrow points to the expected PDFN4 band (predicted molar weight 15 kDa). The membrane was stripped and reprobed with an anti-TUBB antibody (**c’**) and stained with CBB R-250 (**c”**) for protein normalization purposes. Western blotting was performed in six replicates. Statistical evaluation of the IBFC results (**b’’’**) and WB results (**c’’’**) was performed by using ANOVA with Sidak’s post hoc test (α = 0.05).

**Figure 11 biomolecules-13-00996-f011:**
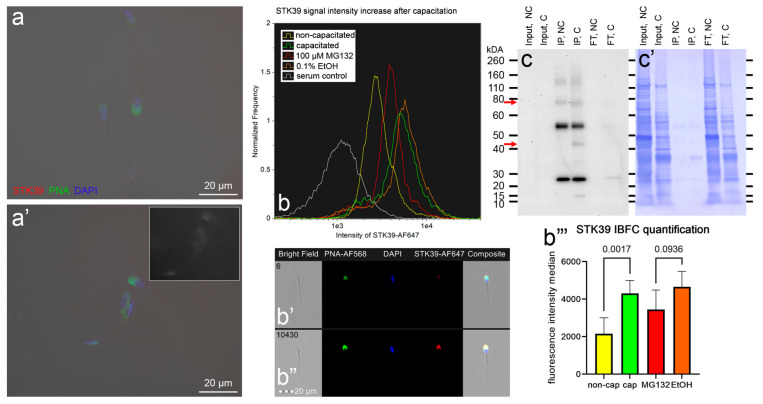
Phenotyping of STK39 in non-capacitated (NC) and in vitro capacitated (IVC) spermatozoa with and without proteasomal inhibition, including a vehicle control. ICC detection of STK39 (red) in NC (**a**) and IVC, non-inhibited (**a’**) spermatozoa; the inset shows STK39 channel only. Spermatozoa were co-stained for acrosomal integrity with peanut agglutinin (PNA lectin, green) and a nuclear stain DAPI (blue). All fluorescence channels were superimposed with the DIC brightfield channel. Scale bars represent 20 µm. The IBFC of formaldehyde fixed and Triton-X-100-permeabilized NC and IVC spermatozoa with or without proteasomal inhibition, including a vehicle control and negative immunolabeling control, normal rabbit serum (**b**). Immunofluorescence images of NC (**b’**) and IVC spermatozoa (**b”**) obtained by using IBFC to confirm the labeling. The IBFC was performed in four replicates with consistent results. Immunoprecipitation of STK39 and WB detection (**c**) in the extracts (input), immunoprecipitates (IP), and non-binding (flow-through, FT) fractions obtained from NC and IVC spermatozoa. The upper red arrow points to the expected STK39 band (predicted molar weight of 63 kDa) and the lower arrow point to the assumed, processed form after the sperm in vitro capacitation. The bands at ~53 kDa and 25 kDa were from the heavy and light chains, respectively, of the STK39 antibody, as anticipated. The membrane was stained with CBB R-250 (**c’**) for protein load estimation. Statistical evaluation of the IBFC results (**b’’’**) was performed by using ANOVA with Sidak’s post hoc test (α = 0.05).

**Table 1 biomolecules-13-00996-t001:** List of significantly different sperm surface proteins between the proteasomally inhibited group and the vehicle group.

Gene Name	UniProtKB Entry	Protein Name	Mass	Up/Down Regulated *	Biological Function in Spermatozoa	Localization in Spermatozoa	Reference
BPHL	F1RWY7 A0A5G2R9J7	Biphenyl hydrolase like	34,119	Down	Detoxification (assumed)	Midpiece, mitochondria	[56,57]
CSE1L	I3LPP4 A0A287BD56 I3L918	Chromosome segregation 1-like protein (Exportin-2) (Importin-alpha re-exporter)	110,649 64,319 109,335	Up	Unknown	Head	[58,59]
GSN	P20305 A0A287A6P1	Gelsolin	84,775 85,683	Down	Actin polymerization during capacitation	Head and flagellum	[60,61]
GSPT1	I3LNK5 A0A287AMP7	G1 to S phase transition protein 1 (Eukaryotic peptide chain release factor GTP-binding subunit ERF3A isoform 1)	69,025 67,988	Down	Unknown	Unknown	[62]
NDUFB7	F1SCH1 A0A4X1V532	NADH dehydrogenase [ubiquinone] 1 beta subcomplex subunit 7	16,458	Up	Oxidative phosphorylation	Midpiece and mitochondria	[63]
NIF3L1	A0A287ASH3 A0A287B238	NGG1 interacting factor 3 like 1	48,992	Up	Unknown	Unknown	[64,65]
PFDN4	A5GHK3 A0A5G2QBU9 A0A8W4FGC6	Prefoldin subunit 4	15,270 14,692 14,712	Down	Molecular chaperone (assumed)	Head	[66,67]
PGLS	A0A287APK0 A0A4X1UE53 A0A287B5V9	6-phosphogluconolactonase EC:3.1.1.31	22,496 23,495 25,491	Up	Gluconeogenesis (assumed)	Unknown	[68]
PPP4C	M3VK32	Serine/threonine-protein phosphatase EC:3.1.3.16	35,080	Up	Cell signaling (assumed)	Unknown	[69,70]
STK39, SPAK	I3LIX4 I3LF98 A0A5G2QZU5	Non-specific serine/threonine protein kinase 39 EC:2.7.11.1	55,762 52,445 63,740	Up	Cell signaling (assumed)	Head	[71,72,73]
STYXL1	A0A4X1UHH1 A0A4X1UJM6 A0A4X1UL86	Serine/threonine/tyrosine interacting like 1	24,717 36,791 35,538	Down	Cell signaling (assumed)	Unknown	[74,75]
TIMM10	A0A286ZVD0	Mitochondrial import inner membrane translocase subunit	10,333	Down	Protein import into mitochondria (assumed)	Midpiece and mitochondria	[76]
TPRG1L	F1RJA8 A0A5G2QRM0	Tumor protein p63 regulated 1 like	25,586 30,487	Up	Unknown	Unknown	
UBXN4	F1S0D8 A0A5G2QFE2 A0A4X1SWJ6	UBX domain protein 4	56,593 52,942 56,593	Down	Unknown	Unknown	

* Up/downregulated in vitro capacitated (IVC) spermatozoa under proteasomal inhibition when compared with IVC vehicle control spermatozoa.

## Data Availability

The data presented in this study are available in the article and Appendix A.

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
