# Peer review of "The Ubiquitin-Proteasome System Participates in Sperm Surface Subproteome Remodeling during Boar Sperm Capacitation"

_biomolecules, 2023, doi:10.3390/biom13060996_

Round 1

Reviewer 1 Report

The study monitors the sperm surface subproteome changes during pig sperm capacitation.

The obtained results mainly confirmed the active role of UPS in the remodeling of the sperm surface subproteome, occuring during sperm capacitation.

In my opinion, the impact of the research is too low to be published in Biomolecules.

In particular, the scientific question proposed in the manuscript should be clearly presented.

Moreover, the description of the results often lacks clarity.

In my opinion, Authors should also connect molecular aspects with their implications in male infertility in humans as well as in animals.

Finally, there are large number of self-citations.

Author Response

We would like to thank the reviewer for their time and feedback. We respect the opinion of this reviewer on the impact of our study. We concur that formulating the scientific question more clearly and connecting the molecular aspects with the implications in male infertility will help to highlight the impact. Based on the reviewer’s advice, we now address both issues in the revised version. We also tried to increase the clarity of the result description. Should this still not be sufficient, we kindly ask the reviewer to provide us with a specific list of suggestions. We believe that self-citations are not a problem if they are used judiciously, which is our case as the present work builds on our previous papers.

Reviewer 2 Report

The present study provides interesting and novel information for the scientific community about the role of the Ubiquitin-Proteasome System on sperm capacitation, a complex event whose molecular bases are not fully known yet.

Please, allow me some suggestions to enhance the quality of this manuscript.

1.      Please, specify the species evaluated in the present study in the title and abstract.

2.      In line 144, describe briefly the capacitating medium.

3.      Why did the authors choose the antibodies anti-CSE1L, anti-PFDN4 and anti-STK39 in section ‘’2.5 Immunofluorescence’’? Specify for a better understanding of the readers.

4.      Line 255 can be removed since the information provided is not new or relevant.

5.      In the statistical section the authors mentioned that four replicates were evaluated in mass spectrometry, however, they should specify that only the data from three working replicates were used for MS data analysis.

6.      Line 343-346, this information has been previously commented on materials and methods, so it can be removed from results.

7.      Line 358, replace the first three most…. By some of the most enriched…. Since the first three most enriched categories (according to Figure 1) were protein folding, protein stabilization, and binding of sperm to zona pellucida. Similarly, the authors made the same mistake in GO cellular and GO molecular functions.

8.      In section ‘’3.4 Localization and dynamics of selected sperm surface UPS protein targets during sperm in vitro capacitation’’, the authors explained that three proteins candidates for UPS modulation were selected for immunolocalization and expression evaluation between the experimental groups, however, they only described CSE1L and STK39. Please, specify what happens with PFDN4.

9.      Line 567, not entirely true, according to Figure 2, the first three most enriched categories were cytoplasm, cytosol, and mitochondrion and not as the authors said mitochondrion, sperm flagellum, and acrosomal vesicle. This sentence and the following should be rewritten.

10.   The last paragraph of the discussion (L739-742) should be added to the conclusions since the conclusions only summarize the results and this is not a conclusion for a scientific article, where the authors should interpret their findings.

Author Response

We would like to thank the reviewer for their praise and feedback; we appreciate their suggestions which we followed to the letter.

  1. Please, specify the species evaluated in the present study in the title and abstract.

Response: This is now included in the title and the abstract.

  1. In line 144, describe briefly the capacitating medium.

Response: A brief description of the capacitation medium is now included.

  1. Why did the authors choose the antibodies anti-CSE1L, anti-PFDN4 and anti-STK39 in section ‘’2.5 Immunofluorescence’’? Specify for a better understanding of the readers.

Response: We chose these antibodies because their target proteins had the highest abundance differences between inhibitor and vehicle controls while also having representatives of up- and downregulated proteins. This is now specified in the Discussion.

  1. Line 255 can be removed since the information provided is not new or relevant.

Response: We removed this line per the reviewer’s suggestion.

  1. In the statistical section the authors mentioned that four replicates were evaluated in mass spectrometry, however, they should specify that only the data from three working replicates were used for MS data analysis.

Response: This is now included in the statistical section, thank you for pointing it out.

  1. Line 343-346, this information has been previously commented on materials and methods, so it can be removed from results.

Response: We agree that the information is redundant, however, we would still like to leave it here for the reader’s convenience.

  1. Line 358, replace the first three most…. By some of the most enriched…. Since the first three most enriched categories (according to Figure 1) were protein folding, protein stabilization, and binding of sperm to zona pellucida. Similarly, the authors made the same mistake in GO cellular and GO molecular functions.

Response: Thank you for spotting this mistake, this happened as the GO enrichment analyses were updated at the last minute. In the previous figures, the GO terms were ordered according to P-values, while the new ones are ordered according to GO term hits. We can see now why reviewer #1 finds the description of the results section confusing and this has been updated.

  1. In section ‘’3.4 Localization and dynamics of selected sperm surface UPS protein targets during sperm in vitro capacitation’’, the authors explained that three proteins candidates for UPS modulation were selected for immunolocalization and expression evaluation between the experimental groups, however, they only described CSE1L and STK39. Please, specify what happens with PFDN4.

Response: The PFDN4 paragraph is now included in the MS. We accidentally deleted it when we applied the Biomolecules template.

  1. Line 567, not entirely true, according to Figure 2, the first three most enriched categories were cytoplasm, cytosol, and mitochondrion and not as the authors said mitochondrion, sperm flagellum, and acrosomal vesicle. This sentence and the following should be rewritten.

Response: The reviewer is indeed correct, as explained earlier, we organized the categories according to P-values at first but then decided to go with the total protein hits. This is now amended in the MS.

  1. The last paragraph of the discussion (L739-742) should be added to the conclusions since the conclusions only summarize the results and this is not a conclusion for a scientific article, where the authors should interpret their findings.

Response: The last paragraph is now a part of the conclusions section. We thank the reviewer once again for pointing out the discrepancies that occurred during the final formatting and Fig 1-4 editing as well as their constructive critique.

Reviewer 3 Report

Comments:

The manuscript has major findings, and it has a good potential for publication; however, in my point of view, the manuscript needs very a few minor modifications as follows:

Comment 1

Please, mention from which sperm species (boar) is used in your work.

Comments 2

Line 46: rather than mention elsewhere, maybe you can change it mentioning the name of the authors that you are citing. 

Comment 3

Line 59: Use the abbreviation sAC instead of SACY for the soluble adenylyl cyclase.

Comment 4

Line 60: The activation of sAC leads to the increase of cAMP, therefore an increase in protein tyrosine phosphorylation will take place, please add this to the sentence.

Comment 5

Line 80: Please write full form of PRKAR1 and AKAP3 proteins in first occurrence.

Comment 6

Line 139: Is not clear which is your non-capacitation conditions, since you are describing that Tyrode medium contains NaHCO3 and CaCl2, please explain or be more specific.

Comment 7

Line 140: Why are you using polyvinyl alcohol on the Tyrore medium composition?

Comment 8

On figure 9C, use cap instead of “capacit”.

Comment 9

Line 519: Use boar instead of pig

Note. This article has almost the same amount of references than a review article.

Author Response

We would like to thank the reviewer for their positive feedback and we appreciate their suggestions which we followed to the letter.

Comment 1

Please, mention from which sperm species (boar) is used in your work.

Response: It is now mentioned in the title and abstract.  

Comments 2

Line 46: rather than mention elsewhere, maybe you can change it mentioning the name of the authors that you are citing.

Response: It is now changed as per the reviewer’s suggestion.

Comment 3

Line 59: Use the abbreviation sAC instead of SACY for the soluble adenylyl cyclase.

Response: It is now changed as per the reviewer’s suggestion.  

Comment 4

Line 60: The activation of sAC leads to the increase of cAMP, therefore an increase in protein tyrosine phosphorylation will take place, please add this to the sentence.

Response: It is now changed as per the reviewer’s suggestion.

Comment 5

Line 80: Please write full form of PRKAR1 and AKAP3 proteins in first occurrence.

Response: It is now changed as per the reviewer’s suggestion.

Comment 6

Line 139: Is not clear which is your non-capacitation conditions, since you are describing that Tyrode medium contains NaHCO3 and CaCl2, please explain or be more specific.

Response: We used modified TL-HEPES-PVA that contained 2 mM NaHCO3 and 2 mM CaCl2 for sperm washing. The non-capacitated sample was directly fixed or frozen for further analysis after the final wash. We updated this information in the MS.

Comment 7

Line 140: Why are you using polyvinyl alcohol on the Tyrore medium composition?

Response: PVA is commonly added to andrology media to prevent spermatozoa from sticking in a non-specific fashion to glass/plasticware or to each other.

Comment 8

On figure 9C, use cap instead of “capacit”.

Response: It is now changed as per the reviewer’s suggestion.

Comment 9

Line 519: Use boar instead of pig

Response: It is now changed as per the reviewer’s suggestion.

Note. This article has almost the same amount of references than a review article.

Response: We thank the reviewer for their recognition.

Round 2

Reviewer 1 Report

The Authors have made substantial improvements in the content. As far as I am concerned, the manuscript is now acceptable to be published.